# Neurons as Monte Carlo Samplers: Bayesian Inference and Learning in Spiking Networks

**Yanping Huang**
University of Washington
huangyp@cs.uw.edu

**Rajesh P.N. Rao**
University of Washington
rao@cs.uw.edu

## Abstract

We propose a spiking network model capable of performing both approximate inference and learning for any hidden Markov model. The lower layer sensory neurons detect noisy measurements of hidden world states. The higher layer neurons with recurrent connections infer a posterior distribution over world states from spike trains generated by sensory neurons. We show how such a neuronal network with synaptic plasticity can implement a form of Bayesian inference similar to Monte Carlo methods such as particle filtering. Each spike in the population of inference neurons represents a sample of a particular hidden world state. The spiking activity across the neural population approximates the posterior distribution of hidden state. The model provides a functional explanation for the Poisson-like noise commonly observed in cortical responses. Uncertainties in spike times provide the necessary variability for sampling during inference. Unlike previous models, the hidden world state is not observed by the sensory neurons, and the temporal dynamics of the hidden state is unknown. We demonstrate how such networks can sequentially learn hidden Markov models using a spike-timing dependent Hebbian learning rule and achieve power-law convergence rates.

## 1   Introduction

Humans are able to routinely estimate unknown world states from ambiguous and noisy stimuli, and anticipate upcoming events by learning the temporal dynamics of relevant states of the world from incomplete knowledge of the environment. For example, when facing an approaching tennis ball, a player must not only estimate the current position of the ball, but also predict its trajectory by inferring the ball's velocity and acceleration before deciding on the next stroke. Tasks such as these can be modeled using a hidden Markov model (HMM), where the relevant states of the world are latent variables $X$ related to sensory observations $Z$ via a likelihood model (determined by the *emission probabilities*). The latent states themselves evolve over time in a Markovian manner, the dynamics being governed by a *transition probabilities*. In these tasks, the optimal way of combining such noisy sensory information is to use Bayesian inference, where the level of uncertainty for each possible state is represented as a probability distribution [1]. Behavioral and neuropsychophysical experiments [2, 3, 4] have suggested that the brain may indeed maintain such a representation and employ Bayesian inference and learning in a great variety of tasks in perception, sensori-motor integration, and sensory adaptation. However, it remains an open question how the brain can sequentially infer the hidden state and learn the dynamics of the environment from the noisy sensory observations.

Several models have been proposed based on populations of neurons to represent probability distribution [5, 6, 7, 8]. These models typically assume a static world state $X$. To get around this limitation, firing-rate models [9, 10] have been proposed to used responses in populations of neurons to represent the time-varying posterior distributions of arbitrary hidden Markov models with discrete states. For the continuous state space, similar models based on line attractor networks [11]

have been introduced for implementing the Kalman filter, which assumes all distributions are Gaussian and the dynamics is linear. Bobrowski et al. [12] proposed a spiking network model that can compute the optimal posterior distribution in continuous time. The limitation of these models is that model parameters (the emission and transition probabilities) are assumed to be known a priori. Deneve [13, 14] proposed a model for inference and learning based on the dynamics of a single neuron. However, the maximum number of world state in her model is limited to two.

In this paper, we explore a neural implementation of HMMs in networks of spiking neurons that perform approximate Bayesian inference similar to the Monte Carlo method of *particle filtering* [15]. We show how the time-varying posterior distribution $P(X_t|Z_{1:t})$ can be directly represented by mean spike counts in sub-populations of neurons. Each model neuron in the neuron population behaves as a coincidence detector, and each spike is viewed as a Monte Carlo sample of a particular world state. At each time step, the probability of a spike in one neuron is shown to approximate the posterior probability of the preferred state encoded by the neuron. Nearby neurons within the same sub-population (analogous to a cortical *column*) encode the same preferred state. The model thus provides a concrete neural implementation of sampling ideas previously suggested in [16, 17, 18, 19, 20]. In addition, we demonstrate how a spike-timing based Hebbian learning rule in our network can implement an online version of the Expectation-Maximization(EM) algorithm to learn the emission and transition matrices of HMMs.

## 2 Review of Hidden Markov Models

For clarity of notation, we briefly review the equations behind a discrete-time "grid-based" Bayesian filter for a hidden Markov model. Let the hidden state be $\{X_k \in \mathbb{X}, k \in \mathbb{N}\}$ with dynamics $X_{k+1} \mid (X_k = x') \sim f(x|x')$, where $f(x|x')$ is the transition probability density, $\mathbb{X}$ is a discrete state space of $X_k$, $\mathbb{N}$ is the set of time steps, and "$\sim$" denotes distributed according to. We focus on estimating $X_k$ by constructing its posterior distribution, based only on noisy measurements or observations $\{Z_k\} \in \mathbb{Z}$ where $\mathbb{Z}$ can be discrete or continuous. $\{Z_k\}$ are conditional independent given $\{X_k\}$ and are governed by the emission probabilities $Z_k \mid (X_k = x) \sim g(z|x)$.

The posterior probability $P(X_k = i|Z_{1:k}) = \omega_{k|k}^i$ may be updated in two stages: a prediction stage (Eq 1) and a measurement update (or correction) stage (Eq 2):

$$P(X_{k+1} = i \mid Z_{1:k}) = \qquad \omega_{k+1|k}^i = \sum_{j=1}^{\mathcal{X}} \omega_{k|k}^j f(x^i|x^j), \tag{1}$$

$$P(X_{k+1} = i \mid Z_{1:k+1}) = \quad \omega_{k+1|k+1}^i = \frac{\omega_{k+1|k}^i g(Z_{k+1}|x^i)}{\sum_{j=1}^{\mathcal{X}} \omega_{k+1|k}^j g(Z_{k+1}|x^j)}. \tag{2}$$

This process is repeated for each time step. These two recursive equations above are the foundation for any exact or approximate solution to Bayesian filtering, including well-known examples such as Kalman filtering when the original continuous state space has been discretized into $\mathcal{X}$ bins.

## 3 Neural Network Model

We now describe the two-layer spiking neural network model we use (depicted in the central panel of Figure 1(a)). The noisy observation $Z_k$ is not directly observed by the network, but sensed through an array of $\mathcal{Z}$ sensory neurons, The lower layer consists of an array of *sensory neurons*, each of which will be activated at time $k$ if the observation $Z_k$ is in the receptive field. The higher layer consists of an array of *inference neurons*, whose activities can be defined as:

$$s(k) = \text{sgn}(\text{a(k)} \times \text{b(k)}) \tag{3}$$

where $s(k)$ describes the binary response of an inference neuron at time $k$, the sign function $sgn(x) = 1$ only when $x > 0$. $a(k)$ represents the sum of neuron's recurrent inputs, which is determined by the recurrent weight matrix $W$ among the inference neurons and the population responses $\mathbf{s}_{k-1}$ from the previous time step. $b(k)$ represents the sum of feedforward inputs, which is determined by the feed-forward weight matrix $M$ as well as the activities in sensory neurons.

Note that Equation 3 defines the output of an abstract inference neuron which acts as a coincidence detector and fires if and only if both recurrent and sensory inputs are received. In the supplementary materials, we show that this abstract model neuron can be implemented using the standard leaky-integrate-and-fire (LIF) neurons used to model cortical neurons.

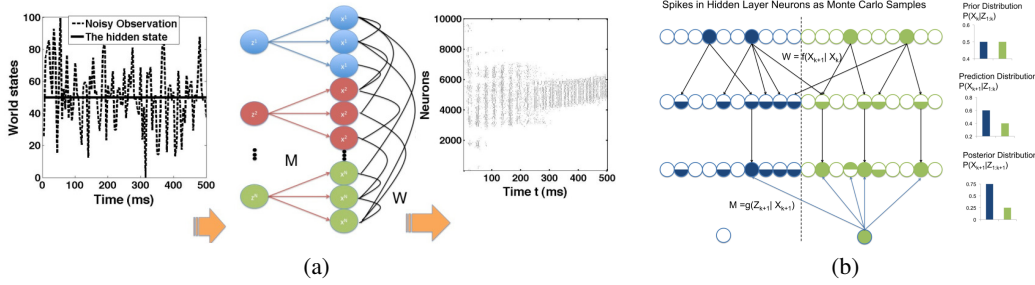

Figure 1: a. Spiking network model for sequential Monte Carlo Bayesian inference. b. Graphical representation of spike distribution propagation

## 3.1 Neural Representation of Probability Distributions

Similar to the idea of grid-based filtering, we first divide the inference neurons into $\mathcal{X}$ sub-populations. $\mathbf{s} = \{s_l^i, i = 1, \ldots \mathcal{X}, l = 1, \ldots, \mathcal{L}\}$. We have $s_l^i(k) = 1$ if there is a spike in the $l$-th neuron of the $i$-th sub-population at time step $k$. Each sub-population of $\mathcal{L}$ neurons share the same preferred world state, there being $\mathcal{X}$ such sub-populations representing each of $\mathcal{X}$ preferred states. One can, for example, view such a neuronal sub-population as a cortical column, within which neurons encode similar features [21].

Figure 1(a) illustrates how our neural network encodes a simple hidden Markov model with $\mathbb{X} = \mathbb{Z} = 1, \ldots, 100$. $X_k = 50$ is a static state and $P(Z_k|X_k)$ is normally distributed. The network utilizes 10,000 neurons for the Monte Carlo approximation, with each state preferred by a sub-population of 100 neurons. At time $k$, the network observe $Z_k$ and the corresponding sensory neuron whose receptive field contains $Z_k$ is activated and sends inputs to the inference neurons. Combining with recurrent inputs from the previous time step, the responses in the inference neurons are updated at each time step. As shown in the raster plot of Figure 1(a), the spikes across the entire inference layer population form a Monte-Carlo approximation to the current posterior distribution:

$$n_{k|k}^i := \sum_{l=1}^{\mathcal{L}} s_l^i(k) \propto \omega_{k|k}^i \tag{4}$$

where $n_{k|k}^i$ is the number of spiking neurons in the $i$th sub-population at time $k$, which can also be regarded as the instantaneous firing rate for sub-population $i$. $N_k = \sum_{i=1}^{\mathcal{X}} n_{k|k}^i$ is the total spike count in the inference layer population. The set $\{n_{k|k}^i\}$ represents the un-normalized conditional probabilities of $X_k$, so that $\hat{P}(X_k = i|Z_{1:k}) = \omega_{k|k}^i = n_{k|k}^i/N_k$.

## 3.2 Bayesian Inference with Stochastic Synaptic Transmission

In this section, we assume the network is given the model parameters in a HMM and there is no learning in connection weights in the network. To implement the prediction Eq 1 in a spiking network, we initialize the recurrent connections between the inference neurons as the transition probabilities: $W_{ij} = f(x^j|x^i)/C_W$, where $C_W$ is a scaling constant. We will discuss how our network learns the HMM parameters from random initial synaptic weights in section 4.

We define the recurrent weight $W_{ij}$ to be the synaptic release probability between the $i$-th neuron sub-population and the $j$-th neuron sub-population in the inference layer. Each neuron that spikes at time step $k$ will randomly evoke, with probability $W_{ij}$, one recurrent excitatory post-synaptic potential (EPSP) at time step $k + 1$, after some network delay. We define the number of recurrent EPSPs received by neuron $l$ in the $j$-th sub-population as $a_l^j$. Thus, $a_l^j$ is the sum of $N_k$ independent (but not identically distributed) Bernoulli trials:

$$a_l^j(k+1) = \sum_{i=1}^{\mathcal{X}} \sum_{l'=1}^{\mathcal{L}} \epsilon_{l'}^i s_{l'}^i(k), \quad \forall l = 1 \ldots \mathcal{L}. \tag{5}$$

where $P(\epsilon_l^i = 1) = W_{ij}$ and $P(\epsilon_l^i = 0) = 1 - W_{ij}$. The sum $a_l^j$ follows the so-called "Poisson binomial" distribution [22] and in the limit approaches the Poisson distribution:

$$P(a_l^j(k+1) \geq 1) \simeq \sum_i W_{ij} n_{k|k}^i = \frac{N_k}{C_W} \omega_{k+1|k}^j \qquad (6)$$

The detailed analysis of the distribution of $a_l^i$ and the proof of equation 6 are provided in the supplementary materials.

The definition of model neuron in Eq 3 indicates that recurrent inputs alone are not strong enough to make the inference neurons fire – these inputs leave the neurons partially activated. We can view these partially activated neurons as the *proposed* samples drawn from the prediction density $P(X_{k+1}|X_k)$. Let $n_{k+1|k}^j$ be the number of proposed samples in $j$-th sub-population, we have

$$\mathrm{E}[n_{k+1|k}^j|\{n_{k|k}^i\}] = \mathcal{L} \sum_{i=1}^{\mathcal{X}} W_{ij}\, n_{k|k}^i = \mathcal{L}\frac{N_k}{C_W}\omega_{k+1|k}^j \propto \mathrm{Var}[n_{k+1|k}^j|\{n_{k|k}^i\}] \qquad (7)$$

Thus, the prediction probability in equation 1 is represented by the expected number of neurons that receive recurrent inputs.

When a new observation $Z_{k+1}$ is received, the network will correct the prediction distribution based on the current observation. Similar to *rejection sampling* used in sequential Monte Carlo algorithms [15], these proposed samples are accepted with a probability proportional to the observation likelihood $P(Z_{k+1}|X_{k+1})$. We assume for simplicity that receptive fields of sensory neurons do not overlap with each other (in the supplementary materials, we discuss the more general overlapping case). Again we define the feedforward weight $M_{ij}$ to be the synaptic release probability between sensory neuron $i$ and inference neurons in the $j$-th sub-population. A spiking sensory neuron $i$ causes an EPSP in a neuron in the $j$-th sub-population with probability $M_{ij}$, which is initialized proportional to the likelihood:

$$P(b_l^i(k+1) \geq 1) = g(Z_{k+1}|x^i)/C_M \qquad (8)$$

where $C_M$ is a scaling constant such that $M_{ij} = g(Z_{k+1} = z^i \mid x^j)/C_M$.

Finally, an inference neuron fires a spike at time $k+1$ if and only if it receives both recurrent and sensory inputs. The corresponding firing probability is then the product of the probabilities of the two inputs: $P(s_l^i(k+1) = 1) = P(a_l^i(k+1) \geq 1)P(b_l^i(k+1) \geq 1)$

Let $n_{k+1|k+1}^i = \sum_{l=1}^{\mathcal{L}} s_l^i(k+1)$ be the number of spikes in $i$-th sub-population at time $k+1$, we have

$$\mathrm{E}[n_{k+1|k+1}^i|\{n_{k|k}^i\}] = \mathcal{L}\frac{N_k}{C_W C_M} P(Z_{k+1}|Z_{1:k})\omega_{k+1|k+1}^i \qquad (9)$$

$$\mathrm{Var}[n_{k+1|k+1}^i|\{n_{k|k}^i\}] \simeq \mathcal{L}\frac{N_k}{C_W C_M} g(Z_{k+1}|x^i)\omega_{k+1|k}^i \qquad (10)$$

Equation 9 ensures that the expected spike distribution at time $k+1$ is a Monte Carlo approximation to the updated posterior probability $P(X_{k+1}|Z_{1:k+1})$. It also determines how many neurons are activated at time $k+1$. To keep the number of spikes at different time steps relatively constant, the scaling constant $C_M, C_W$ and the number of neurons $\mathcal{L}$ could be of the same order of magnitude: for example, $C_W = \mathcal{L} = 10 * N_1$ and $C_M(k+1) = 10 * N_k/N_1$, resulting in a form of divisive inhibition [23]. If the overall neural activity is weak at time $k$, then the global inhibition regulating $M$ is decreased to allow more spikes at time $k+1$. Moreover, approximations in equations 6 and 10 become exact when $\frac{N_k^2}{C_W^2} \to 0$.

### 3.3 Filtering Examples

Figure 1(b) illustrates how the model network implements Bayesian inference with spike samples. The top three rows of circles in the left panel in Figure 1(b) represent the neural activities in the inference neurons, approximating respectively the prior, prediction, and posterior distributions in the right panel. At time $k$, spikes (shown as filled circles) in the posterior population represent the

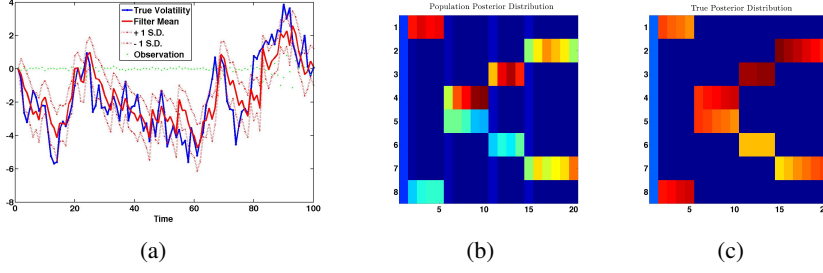

(a)    (b)    (c)

Figure 2: Filtering results for uni-modal (a) and bi-modal posterior distributions ((b) and (c) - see text for details).

distribution $P(X_k|Z_{1:k})$. With recurrent weights $W \propto f(X_{k+1}|X_k)$, spiking neurons send EPSPs to their neighbors and make them partially activated (shown as half-filled circles in the second row). The distribution of partially activated neurons is a Monte-Carlo approximation to the prediction distribution $P(X_{k+1}|Z_{1:k})$. When a new observation $Z_{k+1}$ arrives, the sensory neuron (filled circles the bottom row) whose receptive field contains $Z_{k+1}$ is activated, and sends feedforward EPSPs to the inference neurons using synaptic weights $M = g(Z|X)$. The inference neurons at time $k+1$ fire only if they receive both recurrent and feedforward inputs. With the firing probability proportional to the product of prediction probability $P(X_{k+1}|Z_{1:k})$ and observation likelihood $g(Z_{k+1}|X_{k+1})$, the spike distribution at time $k+1$ (filled circles in the third row) again represents the updated posterior $P(X_{k+1}|Z_{1:k+1})$.

We further tested the filtering results of the proposed neural network with two other example HMMs. The first example is the classic stochastic volatility model, where $\mathbb{X} = \mathbb{Z} = \mathcal{R}$. The transition model of the hidden volatility variable $f(X_{k+1}|X_k) = \mathcal{N}(0.91X_k, 1.0)$, and the emission model of the observed price given volatility is $g(Z_k|X_k) = \mathcal{N}(0, 0.25\exp(X_k))$. The posterior distribution of this model is uni-modal. In simulation we divided $\mathbb{X}$ into 100 bins, and initial spikes $N_1 = 1000$. We plotted the expected volatility with estimated standard deviation from the population posterior distribution in Figure 2(a). We found that the neural network does indeed produce a reasonable estimate of volatility and plausible confidence interval. The second example tests the network's ability to approximate bi-modal posterior distributions by comparing the time varying population posterior distribution with the true one using heat maps (Figures 2(b) and 2(c)). The vertical axis represents the hidden state and the horizontal axis represents time steps. The magnitude of the probability is represented by the color. In this example, $\mathbb{X} = \{1, \ldots, 8\}$ and there are 20 time steps.

## 3.4 Convergence Results and Poisson Variability

In this section, we discuss some convergence results for Bayesian filtering using the proposed spiking network and show our population estimator of the posterior probability is a consistent one. Let $\hat{P}_k^i = \frac{n_{k|k}^i}{N_k}$ be the population estimator of the true posterior probability $P(X_k = i|Z_{1:k})$ at time $k$. Suppose the true distribution is known only at initial time $k = 1$: $\hat{P}_1^i = \omega_{1|1}^i$. We would like to investigate how the mean and variance of $\hat{P}_k^i$ vary over time. We derived the updating equations for mean and variance (see supplementary materials) and found two implications. First, the variance of neural response is roughly proportional to the mean. Thus, rather than representing noise, Poisson variability in the model occurs as a natural consequence of sampling and sparse coding. Second, the variance $\text{Var}[\hat{P}_k^j] \propto 1/N_1$. Therefore $\text{Var}[\hat{P}_k^j] \to 0$ as $N_1 \to \infty$, showing that $\hat{P}_k^j$ is a consistent estimator of $\omega_{k|k}^j$. We tested the above two predictions using numerical experiments on arbitrary HMMs, where we choose $\mathbb{X} = \{1, 2, \ldots 20\}$, $Z_k \sim N(X_k, 5)$, the transition matrix $f(x^j|x^i)$ first uniformly drawn from $[0, 1]$, and then normalized to ensure $\sum_j f(x^j|x^i) = 1$.

In Figures 3(a-c), each data point represents $\text{Var}[\hat{P}_k^j]$ along the vertical axis and $E[\hat{P}_k^j] - E^2[\hat{P}_k^j]$ along the horizontal axis, calculated over 100 trials with the same random transition matrix $f$, and $k = 1, \ldots 10, j = 1, \ldots 20$. The solid lines represent a least squares power law fit to the data: $\text{Var}[\hat{P}_k^j] = C_V * (E[\hat{P}_k^j] - E^2[\hat{P}_k^j])^{C_E}$. For 100 different random transition matrices $f$, the means

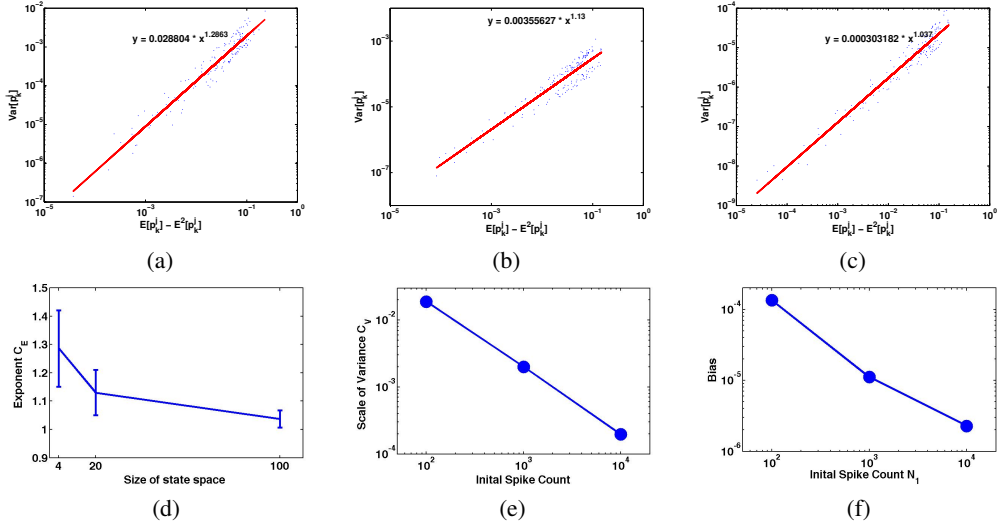

Figure 3: Variance versus Mean of estimator for different initial spike counts

of the exponential term $C_E$ were $1.2863, 1.13$, and $1.037$, with standard deviations $0.13, 0.08$, and $0.03$ respectively, for $N_1 = 100$ and $\mathcal{X} = 4, 20$, and $100$. The mean of $C_E$ continues to approach 1 when $\mathcal{X}$ is increased, as shown in figure 3(d). Since $\text{Var}[\hat{P}_k^j] \propto (E[\hat{P}_k^j] - E^2[\hat{P}_k^j])$ implies $\text{Var}[n_{k|k}^j] \propto E[n_{k|k}^j]$ (see supplementary material for derivation), these results verify the Poisson variability prediction of our neural network.

The term $C_V$ represents the scaling constant for the variance. Figure 3(e) shows that the mean of $C_V$ over 100 different transition matrices $f$ (over 100 different trials with the same $f$) is inversely proportional to initial spike count $N_1$, with power law fit $\hat{C}_V = 1.77 N_1^{-0.9245}$. This indicates that the variance of $\hat{P}_k^j$ converges to 0 if $N_1 \to \infty$. The bias between estimated and true posterior probability can be calculated as:

$$\text{bias}(f) = \frac{1}{\mathcal{X}\mathcal{K}} \sum_{i=1}^{\mathcal{X}} \sum_{k=1}^{\mathcal{K}} (\text{E}[\hat{\text{P}}_k^i] - \omega_{k|k}^i)^2$$

The relationship between the mean of the bias (over 100 different $f$) versus initial count $N_1$ is shown in figure 3(f). We also have an inverse proportionality between bias and $N_1$. Therefore, as the figure shows, for arbitrary $f$, the estimator $\hat{P}_k^j$ is a consistent estimator of $\omega_{k|k}^j$.

## 4   On-line parameter learning

In the previous section, we assumed that the model parameters, i.e., the transition probabilities $f(X_{k+1}|X_k)$ and the emission probabilities $g(Z_k|X_k)$, are known. In this section, we describe how these parameters $\theta = \{f, g\}$ can be learned from noisy observations $\{Z_k\}$. Traditional methods to estimate model parameters are based on the Expectation-Maximization (EM) algorithm, which maximizes the (log) likelihood of the unknown parameters $\log P_\theta(Z_{1:k})$ given a set of observations collected previously. However, such an "off-line" approach is biologically implausible because (1) it requires animals to store all of the observations before learning, and (2) evolutionary pressures dictate that animals update their belief over $\theta$ sequentially any time a new measurement becomes available.

We therefore propose an on-line estimation method where observations are used for updating parameters as they become available and then discarded. We would like to find the parameters $\theta$ that maximize the log likelihood: $\log P_\theta(Z_{1:k}) = \sum_{t=1}^{k} \log P_\theta(Z_t|Z_{t-1})$. Our approach is based on recursively calculating the sufficient statistics of $\theta$ using stochastic approximation algorithms and the

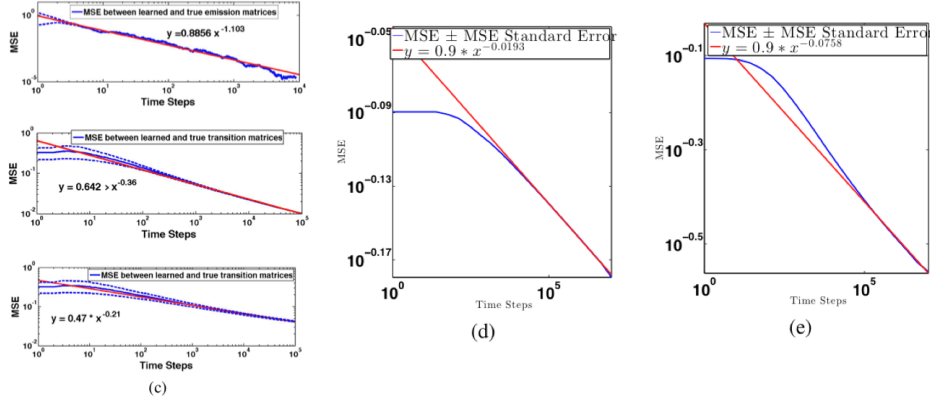

Figure 4: Performance of the Hebbian Learning Rules.

Monte Carlo method, and employs an online EM algorithm obtained by approximating the expected sufficient statistic $\hat{T}(\theta_k)$ using the stochastic approximation (or Robbins-Monoro) procedure. Based on the detailed derivations described in the supplementary materials, we obtain a Hebbian learning rule for updating the synaptic weights based on the pre-synaptic and post-synaptic activities:

$$M_{ij}^k = \gamma_k \frac{n_{k|k}^j}{N_k} \times \frac{\tilde{n}^i(k)}{\sum_i \tilde{n}^i(k)} + (1 - \gamma_k \frac{n_{k|k}^j}{N_k}) \times M_{ij}^{k-1} \quad \text{when } n_{k|k}^j > 0, \tag{11}$$

$$W_{ij}^k = \gamma_k \frac{n_{k-1|k-1}^i}{N_{k-1}} \times \frac{n_{k|k}^j}{N_k} + (1 - \gamma_k \frac{n_{k-1|k-1}^i}{N_{k-1}}) \times W_{ij}^{k-1} \quad \text{when } n_{k-1|k-1}^i > 0, \tag{12}$$

where $\tilde{n}^i(k)$ is the number of pre-synaptic spikes in the i-th sub-population of sensory neurons at time $k$, $\gamma_k$ is the learning rate.

Learning both emission and transition probability matrices at the same time using the online EM algorithm with stochastic approximation is in general very difficult because there are many local minima in the likelihood function. To verify the correctness of our learning algorithms individually, we first divide the learning process into two phases. The first phase involves learning the emission probability $g$ when the hidden world state is stationary, *i.e.*, $W_{ij} = f_{ij} = \delta_{ij}$. This corresponds to learning the observation model of static objects at the center of gaze before learning the dynamics $f$ of objects. After an observation model $g$ is learned, we relax the stationarity constraint, and allow the spiking network to update the recurrent weights $W$ to learn the arbitrary transition probability $f$.

Figure 4 illustrates the performance of learning rules (11) and (12) for a discrete HMM with $\mathcal{X} = 4$ and $\mathcal{Z} = 12$. $X$ and $Z$ values are spaced equally apart: $X \in \{1, \ldots, 4\}$ and $Z \in \{\frac{2}{3}, 1, \frac{4}{3}, \ldots, 4\frac{1}{3}\}$. The transition probability matrix $f$ then involves $4 \times 4 = 16$ parameters and the emission probability matrix $g$ involves $12 \times 4 = 48$ parameters.

In Figure 4(a), we examine the performance of learning rule (11) for the feedforward weights $M^k$, with fixed transition matrix. The true emission probability matrix has the form $g_{\cdot j} = \sim N(x^j, \sigma_Z^2)$. The solid blue curve shows the mean square error (Frobenius norm) $\|M^k - g\|_F = \sqrt{\sum_{ij}(M_{ij}^k - g_{ij})^2}$ between the learned feedforward weights $M^k$ and the true emission probability matrix $g$ over trials with different $g_{\cdot}$. The dotted lines show $\pm 1$ standard deviation for MSE based on 10 different trials. $\sigma_Z$ varied from trial to trial and was drawn uniformly between $0.2$ and $0.4$, representing different levels of observation noises. The initial spike distribution was uniform $n_{0|0}^i = n_{0|0}^j, \forall i, j = 1 \ldots, \mathcal{X}$ and the initial estimate $M_{i,j}^0 = \frac{1}{\mathcal{Z}}$. The learning rate was set to $\gamma_k = \frac{1}{k}$, although a small constant learning rate such as $\gamma_k = 10^{-5}$ also gives rise to similar learning results. A notable feature in Figure 4(a) is that the average MSE exhibits a fast power-law decrease. The red solid line in Figure 4(a) represents the power-law fit to the average MSE: $MSE(k) \propto k^{-1.1}$. Furthermore, the standard deviation of MSE approaches zero as $k$ grows large.

Figure 4(a) thus shows the asymptotic convergence of equation (11) irrespective of the $\sigma_Z$ of the true emission matrix $g$.

We next examined the performance of learning rule 12 for the recurrent weights $W^k$, given the learned emission probability matrix $g$ (the true transition probabilities $f$ are unknown to the network). The initial estimator $W_{ij}^0 = \frac{1}{\mathcal{X}}$. Similarly, Performance was evaluated by calculating the mean square error $\|W^k - f\|_F = \sqrt{\sum_{ij}(W_{ij}^k - f_{ij})^2}$ between the learned recurrent weight $W^k$ and the true $f$. Different randomly chosen transition matrices $f$ were tested. When $\sigma_Z = 0.04$, the observation noise is $\frac{0.04}{1/3} = 12\%$ of the separation between two observed states. Hidden state identification in this case is relatively easy. The red solid line in figure 4(b) represents the power-law fit to the average MSE: $MSE(k) \propto k^{-0.36}$. Similar convergence results can still be obtained for higher $\sigma_Z$, e.g., $\sigma_Z = 0.4$ (figure 4(c)). In this case, hidden state identification is much more difficult as the observation noise is now $1.2$ times the separation between two observed states. This difficulty is reflected in a slower asymptotic convergence rate, with a power-law fit $MSE(k) \propto k^{-0.21}$, as indicated by the red solid line in figure 4(c).

Finally, we show the results for learning both emission and transition matrices simultaneously in figure 4(d,e). In this experiment, the true emission and transition matrices are deterministic, the weight matrices are initialized as the sum of the true one and a uniformly random one: $W_{ij}^0 \propto f_{ij} + \epsilon$ and $M_{ij}^0 \propto g_{ij} + \epsilon$ where $\epsilon$ is a uniform distributed noise between $0$ and $1/N_X$. Although the asymptotic convergence rate for this case is much slower, it still exhibits desired power-law convergences in both $MSE_W(k) \propto k^{-0.02}$ and $MSE_M(k) \propto k^{-0.08}$ over 100 trials starting with different initial weight matrices.

## 5 Discussion

Our model suggests that, contrary to the commonly held view, variability in spiking does not reflect "noise" in the nervous system but captures the animal's uncertainty about the outside world. This suggestion is similar to some previous models [17, 19, 20], including models linking firing rate variability to probabilistic representations [16, 8] but differs in the emphasis on spike-based representations, time-varying inputs, and learning. In our model, a probability distribution over a finite sample space is represented by spike counts in neural sub-populations. Treating spikes as random samples requires that neurons in a pool of identical cells fire independently. This hypothesis is supported by a recent experimental findings [21] that nearby neurons with similar orientation tuning and common inputs show little or no correlation in activity. Our model offers a functional explanation for the existence of such decorrelated neuronal activity in the cortex.

Unlike many previous models of cortical computation, our model treats synaptic transmission between neurons as a stochastic process rather than a deterministic event. This acknowledges the inherent stochastic nature of neurotransmitter release and binding. Synapses between neurons usually have only a small number of vesicles available and a limited number of post-synaptic receptors near the release sites. Recent physiological studies [24] have shown that only 3 NMDA receptors open on average per release during synaptic transmission. These observations lend support to the view espoused by the model that synapses should be treated as probabilistic computational units rather than as simple scalar parameters as assumed in traditional neural network models.

The model for learning we have proposed builds on prior work on online learning [25, 26]. The online algorithm used in our model for estimating HMM parameters involves three levels of approximation. The first level involves performing a stochastic approximation to estimate the expected complete-data sufficient statistics over the joint distribution of all hidden states and observations. Cappe and Moulines [26] showed that under some mild conditions, such an approximation produces a consistent, asymptotically efficient estimator of the true parameters. The second approximation comes from the use of filtered rather than smoothed posterior distributions. Although the convergence reported in the methods section is encouraging, a rigorous proof of convergence remains to be shown. The asymptotic convergence rate using only the filtered distribution is about one third the convergence rate obtained for the algorithms in [25] and [26], where the smoothed distribution is used. The third approximation results from Monte-Carlo sampling of the posterior distribution. As discussed in the methods section, the Monte Carlo approximation converges in the limit of large numbers of particles (spikes).

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
