[Reviews · NeurIPS 2014]

Submitted by Assigned_Reviewer_7

1070: Neurons as Monte Carlo Samplers: Bayesian Inference and Learning in Spiking Networks

The authors study a possible neuronal implementation of learning and inference in a hidden Markov model. As a major novelty, the authors propose that the stochasticity of synaptic transmission is directly involved in the implementation of stochasticity necessary for Monte Carlo sampling.

The neurons used throughout the paper are binary threshold units and not spiking neurons. These binary neurons are able to provide useful insights into how a neuronal network may solve computational problems, but it is important to distinguish between implementations using binary units and spiking neurons. The authors include a short section about spike-based implementation in the appendix, but they do not demonstrate that the spike based implementation is able to perform the same tasks with similar performance. For example, the short term plasticity they propose in the supplementary material to prevent firing after multiple spikes from the recurrent or feed-forward inputs, is not going to help, as STP acts between spikes arriving from at a single synapse not on inputs arriving from a pool of neurons.

Inference: The authors write in the introduction that previous work has been limited to binary word states. However, any discrete state space can be represented by a set of binary variables (just as they do in Fig 2b), hence its is not entirely clear how the paper differs from those previous studies in this aspect.

From the derivation above Eq. 5. and the that C_W~L it follows that the release probability scales with 1/L. However, the release probability in cortical synapses is around 0.5. The authors need to comment on this difference.

During the proof of Eq. 6 the authors take the limit that the spiking activity is sparse. However, it is not clear why this limit is relevant here, especially since sampling is more efficient in the high firing rate limit. The prediction for the Poisson variability also follows from the sparsity assumption and it should be confirmed by simulations.

On Fig. 2a the authors should compare the approximate posterior inferred by the model with the true posterior and not only with the data. They should use a quantitative measure to test the performance of the network (e.g., KL-divergence). They show marginal

An interesting prediction of the theory would be that the sum of the release probabilities at the outgoing synapses of a neuron is constant (C_W) while the same sum for the incoming synapses covaries with the marginal state probability.

Learning:
The learning rule: Eq. 11 and 12 should be replaced by Eq. 31 and 35 of the appendix. Learning is non-local, since it depends on the number of spikes in the (pre and postsynaptic) population (n) and not on the actual spiking of the neuron (s). It is not clear from the data provided whether the learning rule converges to the right value in the case when both the emissions and the hidden dynamics are learned together.
Summary: An interesting paper proposing that stochastic synaptic transmission may be central for implementing probabilistic inference in neuronal circuits through Monte Carlo sampling. The idea itself is interesting, although its possible biological implementation is not clear.

Submitted by Assigned_Reviewer_10

The authors describe a spiking neural network model that can learn hidden Markov models. The output layer of this model describes the posterior distribution of hidden states by having each output neuron depict a sample hidden states. This functionality arises from recurrent connections in the output layer.

This paper is clearly written and original and of high quality. It is difficult to imagine that neurons actually perform this computation, but as theory it is a powerful conjecture and written with such clarity that it provides provable predictions.
Summary: A solid theoretical neuroscience paper that describes neurons as Monte Carlo samplers.

Submitted by Assigned_Reviewer_23

The authors implement approximate inference and learning for a Bayesian Filter for a hidden Markov model in a recurrent neural network. They derive a Hebbian learning rule based on online EM and present experimental tests.

The work seems to be very similar to the recently published article

D. Kappel, B. Nessler, and W. Maass. STDP installs in winner-take-all circuits an online approximation to hidden Markov model learning. PLOS Computational Biology, 10(3):e1003511, 2014.

where learning and inference (Bayesian filtering and full EM, i.e. Baum-Welch algorithm) for HMMs is implemented in spiking neural networks. Both article use very similar network implementations, consisting of recurrently connected WTA subnetworks driven by ffw input.

However, Kappel et al. formulate the problem in continuous time, implement STDP-like learning rules, exponential neural firing rate functions and correct inference resulting in somewhat different learning rules.

In addition to citing Kappel et al. the authors should also comment on the differences between both papers.

I still think it's worthwhile publishing these results, since the approaches are different and in a very similar cases both pieces of work got published:

B. Nessler, M. Pfeiffer, L. Buesing, and W. Maass. Bayesian computation emerges in generic cortical microcircuits through spike-timing-dependent plasticity. PLOS Computational Biology, 9(4):e1003037, 2013.

and

Keck C, Savin C, Lücke J (2012) Feedforward Inhibition and Synaptic Scaling - Two Sides of the Same Coin? PLoS Computational Biology 8: e1002432. doi: 10.1371/journal.pcbi.1002432

In general, the paper is clearly written, although some typos are still in the text (lines 040, 063), the indices in some equations are messed up (Eq. 6 and 7) and the main learning rule (Eq. 12) was copied incorrectly from the suppl. material.

This paper will further facilitate research in the direction of inference and learning based on sequences of patterns in (cortical) neural networks.
Summary: This paper derives strong results on approximate inference and learning of HMM in spiking neural networks, but results for a similar approach have been published recently - limiting its impact.
Author Feedback
Author rebuttal: Reply to Assigned_Reviewer_10:

We are grateful to the reviewer's constructive comments. We would like to describe an equivalent representation of our network which is more biologically plausible. In our network, there are L neurons representing the same hidden state in a subgroup, each of which represents a Bernoulli random variable with success probability proportional to the true posterior probability. When L is large enough, the number of spikes in the same subgroup becomes a Poisson random variable. Thus, each subgroup can be equivalently represented by a Poisson spiking neuron. The inference and learning computations remain the same using this Poisson representation as they only depend on the number of spikes in each subgroup. This implementation trades space for time. For example, it might takes 1s for a Poisson neuron to accumulate 200 spikes while it only takes one time step (10ms using LIF neurons in the suppl. material) to accumulate the same number of spikes in the representation described in the paper.

Reply to Assigned_Reviewer_23:

We thank the reviewer for the additional references and for pointing out the typos. The paper has been revised based on the reviewer's suggestions. We'd like to point out that our approach and the network proposed by Kappel et. al are quite different in many ways:
1. At any HMM time step, the WTA network described in Kappel et. al generates only one sample from the posterior distribution, while in our network of binary neurons, the posterior probabilities are explicitly encoded in the number of spikes in each subgroup. Since the whole distribution rather than a single sample is represented, inference is more accurate. Moreover, the network is able to compute any of the moments of the hidden variable.
2. During inference (forwarding sampling), they assume that the emission and transition models of HMMs are exponentially distributed while our model allows arbitrary HMMs. Moreover, during learning, they only tested their networks using grammar models, which are a special case of HMMs with deterministic observation models.
3. Their rejection sampling method used in HMM learning requires the WTA network to be capable of remembering and replaying the whole input sequence multiple times, thus it is not an online learning method but a batch one. Without the rejection sampling, their online learning method is shown to learn simple grammar models (transitions are deterministic). The network in our paper is able to learn arbitrary HMMs in an online manner.
All in all, the novel contributions of our paper (different from Keppel et. al) are:
1 An online inference algorithm in which the posterior probabilities of hidden states are explicitly represented.
2.Our network is capable of online learning of arbitrary HMM, without any requirements on the memory of past events.

Reply to Assigned_Reviewer_7:

We are thankful for reviewer's detailed comments.

Model Neurons:
We agree that neurons described in our network are binary coincidence detection neurons. The LIF model in the suppl. materials is an example of a biologically plausible implementation. Compartmental models with more sophisticated dendritic architectures might be needed to appropriately implement the contributions from recurrent versus feedforward inputs (for example, Spruston et al, Nat Rev Neurosci. 08). We'd also like to point out the equivalent Poisson representation described in our reply to Reviewer_10, which is biologically plausible.

We apologized for the confusion vis a vis the previous work on "binary" world states. We actually mean that previous work (Deneve 08) is limited to 2-state world states (the number of world states is not more two). Comparisons to other work are listed in the reply to Reviewer_23.

Inference:
In general, the release/binding probability is influenced by a large number of factors. For example, the release probability in rats’ cerebellum is around 0.05 (table 1 of Branco et al, Nat. Rev. Neurosci. 2009), an order of magnitude lesser than the 0.5 probability suggested for the hippocampus. In addition, for simplicity of analysis, our network assumes that the lateral connections is fully connected. However, local connections are highly random by Peters’ rule, “axons make random connections to synaptic targets in the adjacent neuropil, with no local specificity.”(Ohki&Reid Curr Opin Neurobio 07;Kalisman PNAS 05). The 1/L-scale probability is actually the probability of two arbitrary neurons firing together, which is the product between the chance that these two neurons are connected together, and the neurotransmitter release probability (conditioned on the two neurons are connected).

When we take the limit that the spiking activity is sparse, we can model the L neurons in the same subgroup as one Poisson neuron, and the model no longer requires C_W ~ L. We will include the relationships between Poisson variability and sparsity (in terms of L) in the revision.

We will also include KL-divergence to test the performance of our network and the predictions on the sum of release probability in the revision of our paper.

Learning:
The learning rule is local only at the level of subgroup, and not at the level of single binary neurons. When both emission and hidden dynamics are learned together for arbitrary HMMs, we argue that our learning algorithm will converge to the true value due to the power law behavior shown in figure 4. Moreover, in figure 4e, the Frobenius norm between the learned and true recurrent weights is 0.32. For the 16 parameters in a 4x4 matrix, the root mean square error of each element in the matrix reduced from round 0.1 to 0.02 after learning. The computation time for figure 4 only takes a few minutes. We plan to show learning results with more time steps (a few orders of magnitude more) in the final version of this paper.